# Geospatial Assessment of Water-Migration Scenarios in the Context of Sustainable Development Goals (SDGs) 6, 11, and 16

**Nidhi Nagabhatla** [1,2,*] and **Rupal Brahmbhatt** [3]

1   United Nations University, Institute for Water, Health and Environment, Hamilton, ON L8P 0A1, Canada
2   School of Geography and Earth Science, McMaster University, Hamilton, ON L8S 4L8, Canada
3   Department of Geography, University of Northern British Columbia, Prince George, BC V2N 4Z9, Canada; rupal.brahmbhatt@unbc.ca
*   Correspondence: nidhi.nagabhatla@unu.edu

**Abstract:** Communities and countries around the world are gearing up efforts to implement the 2030 Agenda goals and targets. In this paper, the water and migration scenarios are explained with a focus on Sustainable Development Goals (SDGs) 6 (water-related), 11 (urbanization), and 16 (peace and political stability). The study has two phases. The first phase illustrates the application of geospatial data and tools to assess the water-migration interlinkages (nexus) by employing a case study approach. Three case studies, Lake Chad, the Aral Sea region, and the Nile Delta, representing various geographic and socio-political settings, were selected to perform the multitemporal analysis. For this analysis, a mixed toolset framework that combined algorithmic functions of digital image processing, the Landsat sensor data, and applied a geographic information system (GIS) platform was adopted. How water-related events directly or indirectly trigger human migration is described using spatial indicators such as water spread and the extent of urban sprawl. Additionally, the geospatial outputs were analyzed in tandem with the climate variables such as temperature, precipitation data, and socio-economic variables such as population trends and migration patterns. Overall, the three case studies examined how water and climate crisis scenarios influence migration at a local and regional scale. The second phase showcases global-scale analysis based on the Global Conflict Risk Index (GCRI). This indicator reflects on the risks and conflicts with environmental, social, and political aspects and comments on the connection of these dimensions with migration. Together, the two phases of this paper provide an understanding ofthe interplay of water-related events on migration by applying the geospatial assessment and a proxy global index. Additionally, the paper reiterates that such an understanding can serve to establish facts and create evidence to inform sustainable development planning and decision making, particularly with regard to SDGs 6, 11, and 16. Targets such as 6.4 (managing water stress), 6.5 (transboundary challenges) and, 11.B (adaptation and resilience planning) can benefit from the knowledge generated by this geospatial exercise. For example, the high GCRI values for the African region speak to SDG targets 11.B (integrated policies/plans) and 16.7 (decision support systems for peaceful societies). Two key highlights from the synthesis: (a) migration and urbanization are closely interconnected, and (b) the impact of water and climate crisis is comparatively high for rural-urban migration due to the considerable dependence of rural communities on nature-based livelihoods. In conclusion, geospatial analysis is an important tool to study the interlinkages between water and migration. The paper presents a novel perspective toward widening the scope of remote sensing data and GIS toward the implementation of the SDG Agenda.

**Keywords:** water; migration; Landsat; GIS; Sustainable Development Goals 6; 11; 16; interlinkages; scenarios

## 1. Introduction

The global community is in the middle of many environmental burdens, and the water-driven migration context plays a significant role in this discourse. Historically, migration is recognized as a standard approach for adapting to economic crises, political conflicts, or natural disasters at a local, provisional, or national level [1]. Current trends of human migration, particularly forced migration, is considered as a critical developmental challenge [2]. Forced migration refers to the movement of individuals and populations displaced by the impact of development projects such as the construction of hydropower systems such as dams, natural and environmental stressors, chemical and or nuclear disasters, conflict, and situations related to water, food, and energy crisis. These processes may cause the loss of livelihood and income generation opportunity, damage of assets including habitation, and as a result, migration and displacement, either voluntary or forced [3,4]. Up-to-date and comprehensive knowledge of the direct and indirect drivers of migration can serve to create evidence for designing future migration policies and actions. The current literature on human migration has mostly examined the social and humanitarian context (i.e., the post-migration response). Lechner and Renault commentary on migration endorse that the existing studies are rich in qualitative assessments and descriptive narratives. However, quantitative information on water and climate crisis influenced migration is under reflected within the existing literature and information [5].

The International Panel on Climate Change (IPCC, 2007) report reflects on the influence of climate change on local and regional populations and how migration is increasing due to droughts, water shortages, and coastal flooding [6]. Various future estimates point to an escalating vulnerability of the human community and note that in the recent future billions will experience water scarcity or will be impacted by coastal flooding, and a further million may face food shortages [7]. The International Organization for Migration (IOM, 2011) predicts that in the coming 20 to 30 years (i.e., by 2050,) over more than 200 million people will have to migrate due to the impacts of climate change, ecological degradation, and natural disasters [8]. According to the UN World Water Development Report (WWDR, 2019), 50 liters of water per day per capita is the essential water need. Any deviations from this value of basic water need, whether in developed or developing countries, arise due to accessibility availability and water quality-related challenges [9,10]. The quality and quantity of drinking water can also act as a trigger to conflict and strife in vulnerable populations, particularly in water-stressed regions [9]. Therefore, regular monitoring and sustainable management of water resources, and smart and integrated planning of cities and habitations is a priority for safeguarding human well-being, peace, and political stability. The connection between the water and migration scenarios with SDGs 6 (water) and 11 (urbanization), and 16 (institutions, and policies) is obvious.

The explanation by Werner et al. on climate and environmental migration illustrates increasing water scarcity and water stress as triggers for migration, especially in densely populated regions with limited freshwater resources [11]. Environmental changes and its likely influences are no longer sporadic or localized. Instead, humans are at the center of alterations in ecological trajectories and the resulting spillovers (i.e., impacts of such modifications are apparent). The new and emerging pathways of migration are influenced by a variety of direct and indirect drivers. For instance, the increasing trend of migration or forced displacement after prolonged water-related events (e.g., droughts and floods) are noted in many parts of the world [12,13]. Sunjic claims that climate change influences the mobility of more than five million people/year and as such, individuals and groups living in vulnerable situations, where these communities will continue to be adversely impacted [14]. Therefore, it is important to understand how the biophysical characteristics of water and related ecosystems vary with time, either as a result of anthropogenic interferences or due to the impact of nature-based triggers and how these variations influence the socioeconomic structure of communities and states.

Identifying data and information gaps is a critical need toward a comprehensive understanding of the water-migration interlinkages and so is the inquiry of various drivers and dimensions that affect migration trends and patterns. Toward this need, geospatial tools and techniques allow the integration of geographically referenced data, combined with ground knowledge, to develop relational databases.

Such aggregations can facilitate the display of the complexity of interactions and interlinkages in a simple format [15]. It is commonly agreed that geospatial tools demonstrate the potential to enhance our understanding of the various and diverse drivers of land-use changes [16]. In turn, changing land use, infrastructure development, and rapid urbanization are manifestations of socio-economic and socio-political variables such as government policies, land and water tenure guidelines, inequality, wealth and power dynamics, market mechanisms, and customary norms, among others. These dimensions are often mentioned in the migration literature as driving factors of human mobility; however, these have not often been investigated alongside the spatial and temporal dynamics of land and water resources [17].

Remote sensing images and geographic information systems (GISs) data, tool, and platforms demonstrate the potential to examine land and water use trends and patterns, and potential toward the 'integration of' and 'correlation with' socio-economic aspects (primary and secondary data, e.g., migration and population trends). Such integration, in turn, can allow us to understand the interface of land and water dynamics with migration. Mapping the vegetation and water changes remains a vital aspect of planning the integrated assessments. For instance, the challenges connected with land and water crises (i.e., issues of access and availability), can be explained through the changing area and water use/spread in a time series data. Examining the interlinkages is also significant to assess out-migration studies, primarily those exploring rural-urban migration pathways [18]. For example, change in land use for agriculture production in a space-time continuum relate to mobility patterns of people to a good extent, more so, for communities and individuals involved in the rural sector Understanding this correlation, in turn, can assist in explaining urbanization flow to adjoining cities and urban centers (i.e., rural to urban migration).. To this, Geogheagan et al. presented an excellent example of applying GIS toward examining the effects of land-use changes and their impact on the economic value of natural resources. Additionally, the researchers commented on the use of models that combined remote sensing observations with ground-based social data to cast the probability of future land development [16]. In the 1980s, Turner's explanations of geospatial analysis as a mechanism to outline and apply surrogate indicators for natural resource management studies [19] followed research from various disciplines exploring the application of GIS and satellite data for the assessment of temporal trends and spatial patterns. This scholarship work presents the prospect of scaling this application to assess the water-migration interlinkages.

Considering the above arguments, this paper explains the water and migration scenarios by applying a set of geospatial data and tools. The study was divided into two phases. The first phase adopted a multitemporal analysis approach to analyze the land use and cover change at a local/regional scale for three case studies. The second phase illustrated a global analysis based on the Global Conflict Risk Index (GCRI). The overarching objective of this study was to propose a scalable and interoperable framework (for water-driven migration assessment) that integrates remote sensing data and GIS tools, techniques, and indicators.

Phase 1: The selected case studies were Lake Chad, the Aral Sea region, and the Nile Delta as they represent various geographic and socio-political settings. The geospatial assessment approach for this phase takes account of a mixed toolset framework that combines the algorithmic functions of digital image processing, the Landsat sensor, and the geographic information system platform. It derives spatial indicators such as water spread and the extent of urban sprawl to comment on how water-related events directly or indirectly trigger migration. Furthermore, the geospatial outputs were analyzed in tandem with the climate variables such as temperature, precipitation data, and socio-economic variables such as population and migration data to examine how water and climate crisis scenarios connect with migration trends. This phase also demonstrates the application of GIS and remote sensing data to explain the water and migration scenarios related to water quality, quantity (availability and access), and water extreme (mainly droughts and floods) episodes.

Phase 2: The second phase showcases global-scale analysis based on GCRI. This indicator reflects the risks and conflicts with the environmental, social, and political aspects and explains the connection of these dimensions with the outcomes of conflict such as migration.

Overall, the two-phased analysis integrated the targets and goals specific to SDGs 6, 11, and 16 related to water stress, urbanization, and institutional aspects, respectively.

The sub-objectives of the study include the following:

- To present and analyze appropriate, fitting, and representative case studies that illustrate the water-migration scenarios by applying a mixed set of geospatial tools and techniques (e.g., multi-temporal analysis).
- To assess the impacts of the water crisis on migration within the context of SDG 6, 11, and 16.
- To provide an overview of global indicators such as GCRI using a GIS domain toward the facilitation of long-term sustainability and an enhanced understanding of the water-migration interlinkages (nexus) and how these interlinkages interface with risks and conflicts.

## 2. Study Area

Migration has seen a significant rise in different places worldwide. The selected case studies (a) Lake Chad; (b) the lower Nile Delta; and (c) the Aral Sea region capture the diversity in the water-migration interactions in various geographic settings (Figure 1).

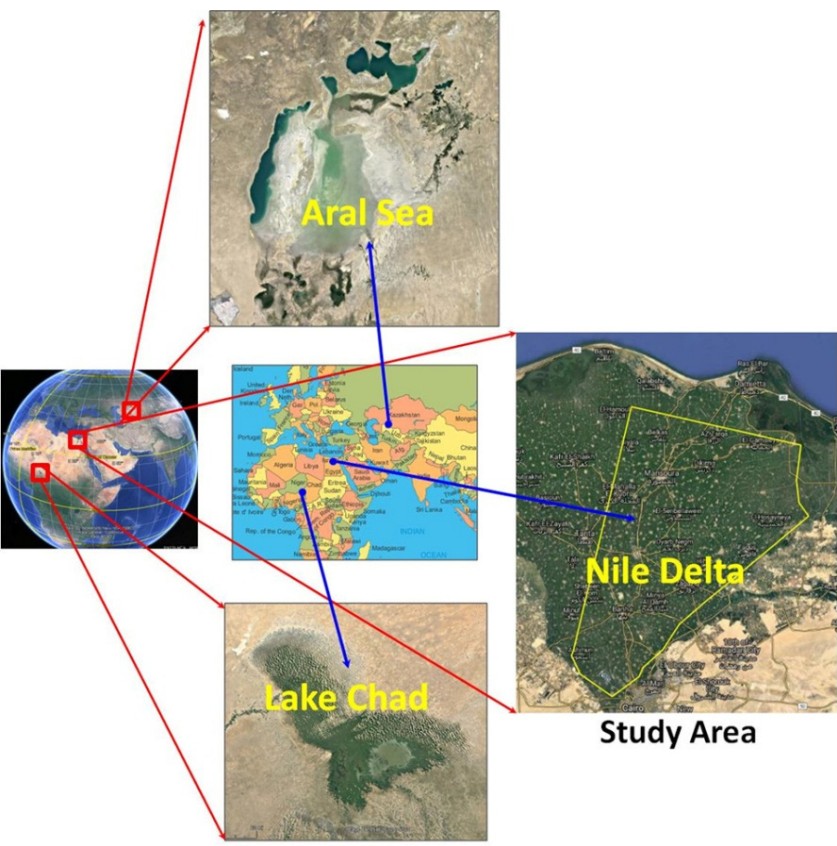

**Figure 1.** Location of the case study sites in Africa and Central Asia.

Lake Chad is in the southern margin of the Sahara Desert. It constitutes several water bodies including two basins and a permanent water pool facing the Chari Delta in the south of the basin [20]. The southern basin is fed directly by the inflow of the Chari/Logone River. The northern part of the basin receives a small flow from the Yobe River [21]. The lake is considered as a closed water system (no outflow). As the lake is in a semi-arid region in the south and arid in the north, the water crisis (due

to persistent droughts) remains a continuing challenge. Additionally, like any other transboundary water system, in the past decades, the lake has experienced its share of socio-economic complexity, crises, and conflicts related to water [22,23]. In the 1960s, it was the sixth-largest lake in the world, with the open water area spreading approximately 25,000 km$^2$ in the four states of the northern Africa region, namely Chad, Cameroon, Niger, and Nigeria. It served as a vital source of water, providing for over 30 million people in the Sahel and activities including farming, fishing, and pastoral cultural practices [24]. Since the 1960s, the lake has shrunk drastically, wherein the spread of 22,000 km$^2$ in the 1980s reduced to about 1304 km$^2$. Scientific investigations have mostly attributed this change to droughts and an increase in irrigation withdrawals [24].

Inhabitants of the lake basin (with the highest population in Nigeria followed by Cameroon, Niger, and Chad) depend on it for water provisioning and income generation through fisheries, agriculture, and other activities [25]. The water recession in the second half of the 20th century triggered conflicts related to food security, poverty, and migration [26,27]. Although a few studies and experts have referenced droughts in the above incidents, these events did contribute to modify the hydrodynamics of the lake basin and the adjoining floodplains. Limited attention was given to understand the larger impacts on the communities and populations. For example, the loss of livelihood and income, as well as human displacement, are among the relatively under-investigated aspects. A total of 2111 (1435 in Nigeria, 541 in Cameroon, and 135 in Chad) locations host populations affected/displaced in the Lake Chad region. Of these trends, over 60% of displacement occurred in 2014 and 25% in 2015 [28]. Lake Chad also offers a typical case to analyze water crisis and migration within the SDG context (targets 6.4, 6.5, and 16.7)

The second case study, the Nile Delta, is a region around the Damietta tributary of the Nile. The region was selected as the area of interest for the multitemporal dynamics. This delta region is a vital feature of the longest river in the world, the Nile (6760 km), which crosses ten countries and feeds Lake Victoria (African Great Lake) [29]. The Blue Nile region begins at Lake Tana (Ethiopia) and joins the White Nile (originating in Burundi) in Khartoum (Sudan). From the north to south, the delta spreads for 160 km. Furthermore, the delta spans a 240 km coastline from the west to east. Egypt has historically (>100 years) witnessed both internal and international migration and internal displacement (Table A1). The population density of the urban centers in the delta region averages 1000/km2 with the largest city, Alexandria, inhabited by over 4.5 million people. Other urban centers have also reported dynamic urban growth in past years such as Shubra al Khaymah, Port Said, El-Mahalla El-Kubra, El Mansura, Tanta, and Zagazig. Researchers and media interfaces have briefly reported on the water crisis and the movement of people in the region [30,31]. The steady rise of urbanization in this region is a key facet in the multitude of migration-related impacts.

The third case study was the Aral Sea lake region, which is located east of the Caspian Sea, between Uzbekistan and Kazakhstan in Central Asia. In this region, the quick and sharp rise in urban sprawl, shifting settlement zones, and flux in water resources has driven the economy to income deprivation from civic activities. In short, rural out-migration is one notable outcome. Thus, the Aral Sea provides a compelling case to examine the consequences of water-focused developmental interventions and their impacts on the human population, settlements, and migration (mainly rural out-migration). At one point in time, the Aral Sea was the world's fourth-largest freshwater lake, feeding two large rivers in Central Asia: Amu Darya (2580 km long), originating in Tajikistan with channels from northeastern Afghanistan, and Syr Darya (2220 km long) originating in Kyrgyzstan. In the 1950s, the region was rich in biodiversity and aquatic flora and fauna (fish, birds, and other wildlife), and the aquatic ecosystem had a water volume exceeding 1090 km$^3$ (1 km$^3$ = 1 billion m$^3$ = BCM). During the 1960s, its surface area would spread more than 67,500 km$^2$ [32]. The water level in the lake basin exhibits seasonal fluctuations between 50–53 m above sea level [33,34]. The impacts of climate change (high frequency of droughts) is exacerbated by decades of ecological degradation, short term planning, and geopolitical complexities, which in turn has produced various outcomes such as the movement of people and communities (migration), particularly impacting vulnerable groups such as farmers and fishers.

## 3. Data and Methods

Figure 2 shows the schematic framework of the methodology applied in the two-phased analysis. The collective set of tools applied to investigate the spatial and temporal dynamics of the selected region includes digital images from medium resolution satellite sensors: Landsat MSS and TM. Around 115 satellite images were used to extract features such as vegetation and urban coverage by applying indices-based classification and visualization techniques. Table A2 provides the list of the set of images with dates, and the number of images used for each case study. The images were joined to form a mosaic and to extract the area of interest (AOI) for the multi-temporal analysis. Toward deciding the AOI, a set of parameters such as visual clarity cloud-free scenes and available time-series data was considered. A total of 50 satellite scenes were utilized for the analysis of Lake Chad (Central Africa), 15 for the selected Nile Delta region (Egypt), and 60 for the Aral Sea (Central Asia).

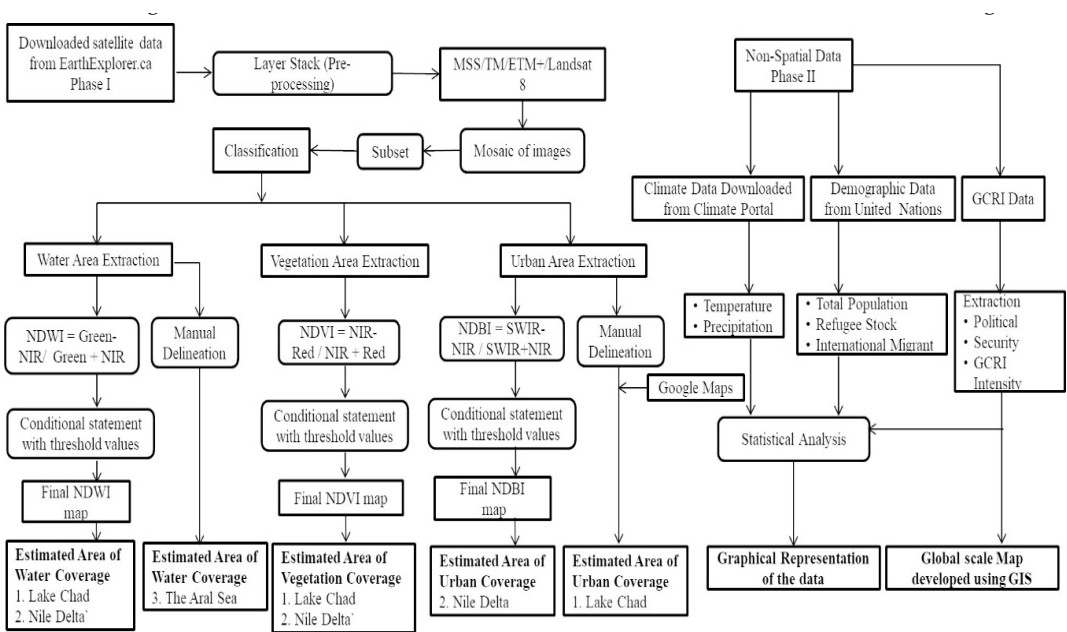

**Figure 2.** Schematic flow of the methodological framework applied in the study for Phase 1 and Phase 2 of the analysis.

A set of indices and pre-set algorithms, namely, normalized differential vegetation index (NDVI), normalized differential water index (NDWI), and normalized differential built-up index (NDBI), along with visual interpretation techniques including color, texture, patterns, and proximity were considered to delineate specific features connected with the objectives of the study. In addition to the satellite data, climate data (temperature and precipitation) for the period of 1901–2015 were integrated into the analysis. The United Nations data portal (https://data.un.org/) was used to extract the demographic data to draw the population trends (1927 to 2017). The international migrant stock, refugees, asylum seekers data were accessed from the United Nations Global Migration Database (UNGMD).

Phase 1 of the Geospatial Analysis: For this, the digital image processing of the remote sensing data using band ratio and visual interpretation was applied [35,36]. Surface water cover was estimated using the NDWI [37] as the index fits well with the Landsat Multispectral Scanner (MSS) image data, and with other sensor systems in applications where the measurement of the extent of open water is needed [38–42]. The threshold value of the NDWI pixel value was assessed [37]: 0 represents the water pixel, whereas values ≤ 0 were mapped as non-water pixels. Additionally, >0 threshold value was used to delineate the surface water area. Ceola and colleagues introduced the new approach to identify anthropogenic impacts on water resources by using satellite-derived nighttime light [43]. NDVI was used for mapping vegetation cover and change detection [44–46]. The threshold varied between −1 (represent the sparse vegetation) to 1 (represent dense/healthy vegetation). The urban area

was mapped using NDBI based on the spectral signature of the built-up area in the SWiR and NIR regions. The overall accuracy of the mapped built up was measured at 92.6 percent, which was in line with similar studies [47]. NDBI values close to 0 were woodland/farmland pixels, whereas negative values indicated water bodies. The positive NDBI value was for built-up areas that were separated from the remaining land use type. The threshold of the index values varied with the satellite data [47].

The Google Earth platform was used as an additional resource to delineate the vector layer and support the analysis and the verification process. The platform was also referred for manual correction and to fill the information gaps due to missing data in the Landsat mosaic, especially for the Lake Chad region. The NDBI approach did not apply well in that case study as the expanse of the built-up area was very small, mostly a small spread of villages (huts). Overall, the combination of multi-temporal analysis in a geospatial medium and case study approach was adopted to depict variations in the water crisis scenarios acting as direct and indirect drivers of migration. For each case study, demographic and climatological data were used to complement the analysis and to comment on the interlinkages derived from multitemporal analysis, climate variability, and plotting of migration trends. This analysis also examines the correlation of urban growth with the temporal dynamics of land use and water spread and climatic variability. The relationship with SDG 6, 11 and 16 goals, mainly targets 6.4, 11.5, 11.6, and 16.7 is intertwined for each of the case studies, and is explained along with the results.

Phase 2 of the Geospatial Analysis: This part was based on the GCRI (Global Conflict Risk Index), which is a quantitative conflict risk index developed by the European External Action Service (EEAS) to enhance conflict prevention capacities in close partnership with the European Commission, and the European states. It is designed based on open-source data, with an aim to provide input to an early warning framework and toward outlining conflict early warning systems [48]. The index represents the statistical risk of violent conflicts based on quantitative indicators from open sources, (i.e., 24 variables in five dimensions, namely social, economic, security, political, geographical/environmental). The analysis employed a statistical regression model to calculate the probability and intensity of violent conflicts using a scale of 0–10 (0 is equivalent to no conflict). The approach was derived based on the assumption that understanding the pattern of violent conflicts in a country allows for the assessment of the efforts and interventions required for planning stability and sustainability. In this study, a geospatial platform was used to display various dimensions of GCRI on a global scale, and to comment on how risks and conflicts related to migration.

## 4. Results

### 4.1. Phase 1: The Water-Migration Nexus – the Case Study Approach

The three case studies explained in this phase aimed to demonstrate the role of geospatial data, tools, techniques, and indicators toward an enhanced understanding of the water-migration scenarios. The specific arguments explained in each of the case studies is to assist with the unpacking of the nuanced interlinkages that apply to water - migration nexus.

Case Study 1: Lake Chad, Africa

A large part of the lake lies in Chad (55%), followed by Nigeria (~20%), Niger (~15%), and Cameroon (~10%). The lake basin is mostly rural and hosts many villages (temporary and thatched shelters as observed in the Google Earth images). The open water is separated into two parts by a shallow zone referred to as the "Great Barrier" [24]. The surface water share of the Lake Chad basin in Niger and Nigeria is small compared to that of Chad and Cameroon [49]. The geospatial multi-temporal analysis and historical chronology both reflect on the degradation of this aquatic ecosystem. In 1973, the surface water spread of the lake was approximately 14,300 km$^2$; this had reduced to 1300 km$^2$ by 2017 (see Figure 3). Following, in 2017, the permanent or seasonal marshes covered most of the lake basin area. Post-2000s, the northern region of the lake largely dried up and continued to shrink in the surface water coverage (Figure 3a,b). Restoration interventions did help restore it temporarily to some

extent and to sustain the leftover surface water. In 2017, the mean area in the southern basin measured 1450 km² [50].

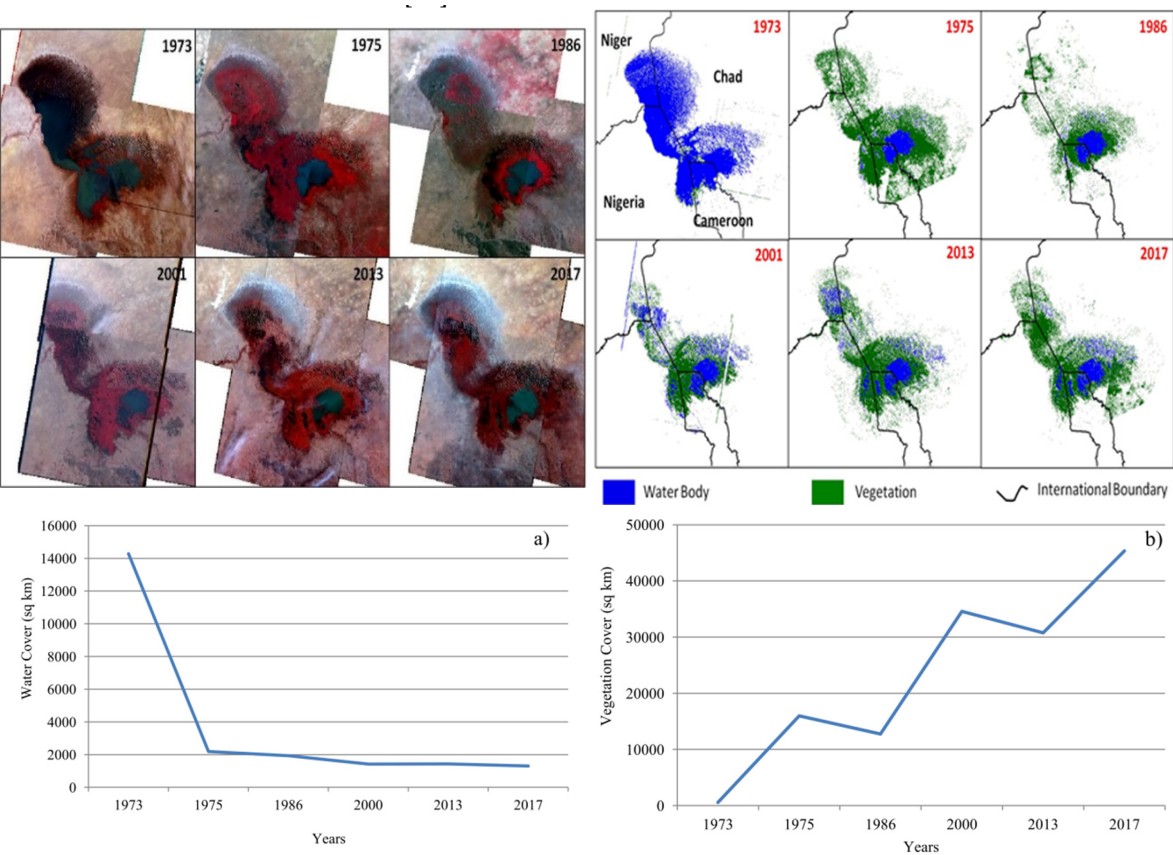

**Figure 3.** The remote sensing (Landsat data overview) and the derived vegetation and water spread in Lake Chad (1973–2017). Temporal trends (**a**) surface water spread (km²) and (**b**) vegetation cover spread (km²), plotted based on spatial statistics.

The temporal rainfall variations (2013–2017) add to the explanation derived from the multitemporal analysis, as does the trend analysis of the precipitation changes and population growth patterns (Figure 4a,b). For example, the marsh vegetation around the water spread reported improvement in 1975 and decline a decade later. From 2001 onward, the native vegetation intermittently reflected signs of recovery attributed to a mix of factors such as climatic variations, restoration efforts, and an emphasis on the sustainable use of lake resources for provisioning services (e.g., fisheries, agriculture). In years with regular rainfall intensity, the surface water and the surrounding vegetation of the basin noted signs of restoration.

The surface area shrinking trend is explicitly demonstrated by the multi-temporal analysis. It is anticipated that pressure from conflicting and competing use of lake resources is further exacerbated by inflow (discharge of the rivers in the lake) and precipitation variability. The release of the Chari/Logone River has almost decreased by 50 percent over 40 years [27]. Additionally, an increasing temperature trend has influenced the evapotranspiration balance of the region [24]. Gao et al.'s analysis revealed how persisting drier conditions and droughts, along with an increase in irrigation demands, have risen four-fold between 1983–1994 [24]. Consequently, this has led to aggravating pressure on water availability and allocation. A study on climatological and river discharge investigation at the basin scale with a focus on the southern region of Lake Chad has illustrated the impacts of precipitation decline since 1985 [51].

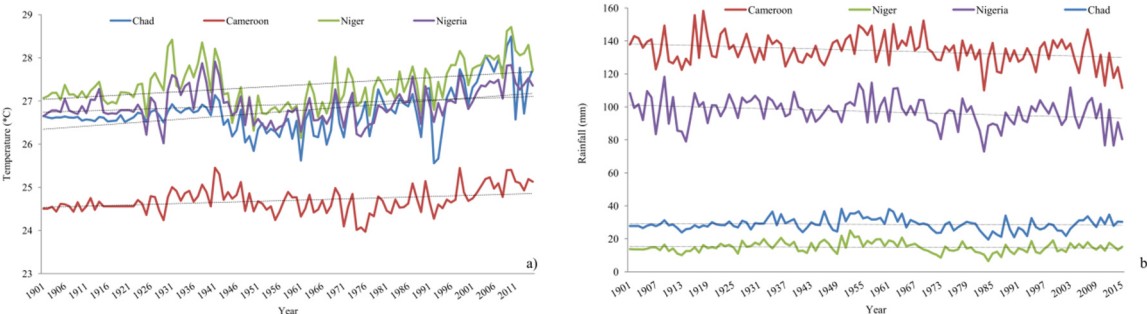

**Figure 4.** Variability in the (**a**) annual mean temperature and (**b**) rainfall for Chad, Cameroon, Niger, and Nigeria around the Lake Chad basin.

The spatial distribution of the habitation and settlements (produced by buffering 100 km area around the lake boundary and taking references from Google Earth data) around the lake derived using Landsat is shown in Figure 5. It shows villages or areas with thatched areas/huts as red dots. The states of Chad and Niger are landlocked countries that are water-dependent (on rainfall, inland water bodies, and wetlands) for food production and other provisioning needs. The population growth is increasing in all states surrounding the lake basin (Figure 6a). Settlements developed on ancient dunes surrounding the seasonal water bodies facilitate livelihood and income generation for resource-dependent communities. The habitation zone mapping for 2017 is a useful indicator for reflecting the water use needs for both resident and migrant communities. As more migrants settle in the basin, water use arrangements and allocation policies will need further attention. The interface of the current settings of the Lake Chad region with the goals and targets of SDGs 6 (water-related), 11 (managing sustainable habitations), and 16 (polices and institutions) seem relevant.

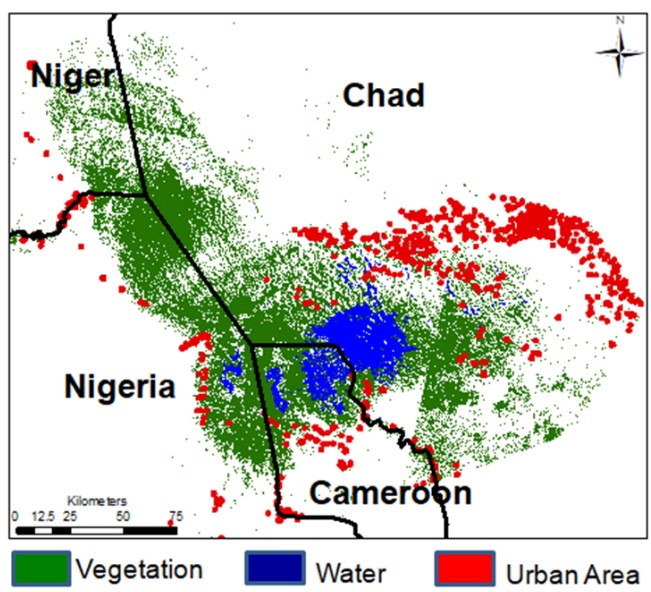

**Figure 5.** Habitation spread in the Lake Chad region (2017) derived using remote sensing data and GIS tools.

The Chad and Nigeria regions reported high migration during 2005–2017, both internally and internationally (Figure 6b). Quantitative estimations of international migration in this region may not be as accurate due to limited data collection. Although all states surrounding the basin recorded high migration flows during 1990–2017, the Chad region showed the highest absolute number of international migrants (recorded as refugees and asylum seekers) when compared to Cameroon, Niger, and Nigeria. A significant portion of the open water surface of the Lake has dried up in the past

decades, while farmers and cattle herders have moved toward greener areas where they compete for land resources with host communities. Others have moved to Kano, Abuja, Lagos, and other large cities in the African region looking for unskilled jobs or have remained jobless. Reports state that over 2.5 million people have fled their homes due to the ongoing political conflicts mainly concerning conflicts by Boko Haram (an extremist group adhering to certain religious beliefs spreading in West Africa). Some studies have also reflected on how water and food insecurity scenarios either fuel existing political conflicts or are leveraged by such conflicts, toward a substantial bearing on migration [52,53].

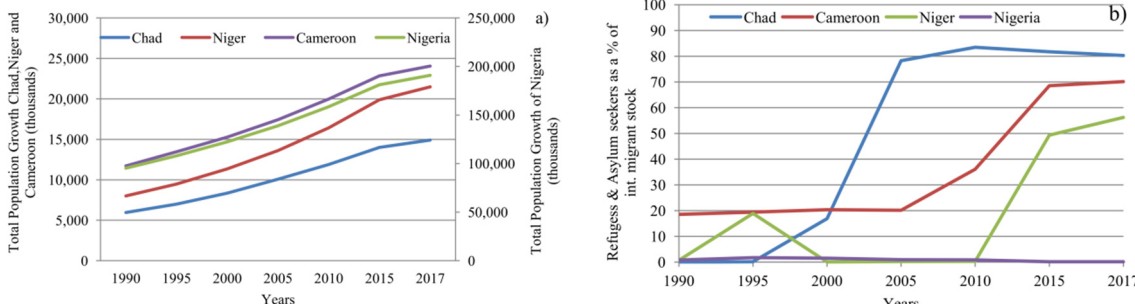

**Figure 6.** Trend (**a**) population growth and (**b**) migration flow(refugees) in the Lake Chad region (Cameroon, Chad, Niger, and Nigeria).

Case Study 2: The Nile Delta Region, Africa

The Nile Delta region was analyzed using remote sensing data (Landsat images from 1970–2017) of nearly five decades. The Area of Interest (AOI), Damietta region, is located close to Cairo. The mapping and multi-temporal analysis displayed the surface water spread, the urban growth, and the change in vegetation for the Damietta region (Figure 7). The shrinking of the surface water spread in Damietta since the 1970s is one critical observation from the geospatial analysis. In the 1970s and part of the 1980s, the river had a regular surface water flow and healthy aquatic ecosystems in the adjacent sub-basin. In the 1990s, the water flow spread was reduced and declined thereon. The spatial patterns, as shown in Figure 7a,b, point to the signs of shrinking vegetation and water cover in the previous decades and align with the study by Ahmed et al. [54], who recorded the shrinking of wetlands or aquatic ecosystems in the delta region. In this assessment, it was noted that the areal spread of vegetation in 2017 was almost half of the baseline noted in 1972, and urbanization has been on a dramatic rise (Figure 7c).

The delta region is a popular habitation site and tourist destination and is rich in biodiversity that serves as a habitat for many floral and faunal species. Urban coverage estimated using satellite image processing shows the significant spread in the past four decades with the existing urban center, either expanding and/or shifting toward the coastal zone. The effects of climate variability examined by annual mean historical rainfall trends and temperature variations from 1901–2015 (Figure 8) corroborates the climate change impacts in the supply-demand dynamics of the water sector in the Nile region, as explained in other studies [55]. The vegetation cover, as well as the water spread in the region, has been influenced by climate change impacts, and in turn, these changes affect the socioeconomics and result in migration and the expansion of human settlements, mostly in urban areas. However, these interlinkages may not be linear all the time [56]. The Nile region experienced heavy rains in 1994, resulting in flash floods, and the loss of assets and a displacement of over 160,000 people. In 2010, flooding affected another 3500 people (12 lives lost), as shown in the Climate Change Information Fact Sheet of Egypt drawn by USAID in 2015.

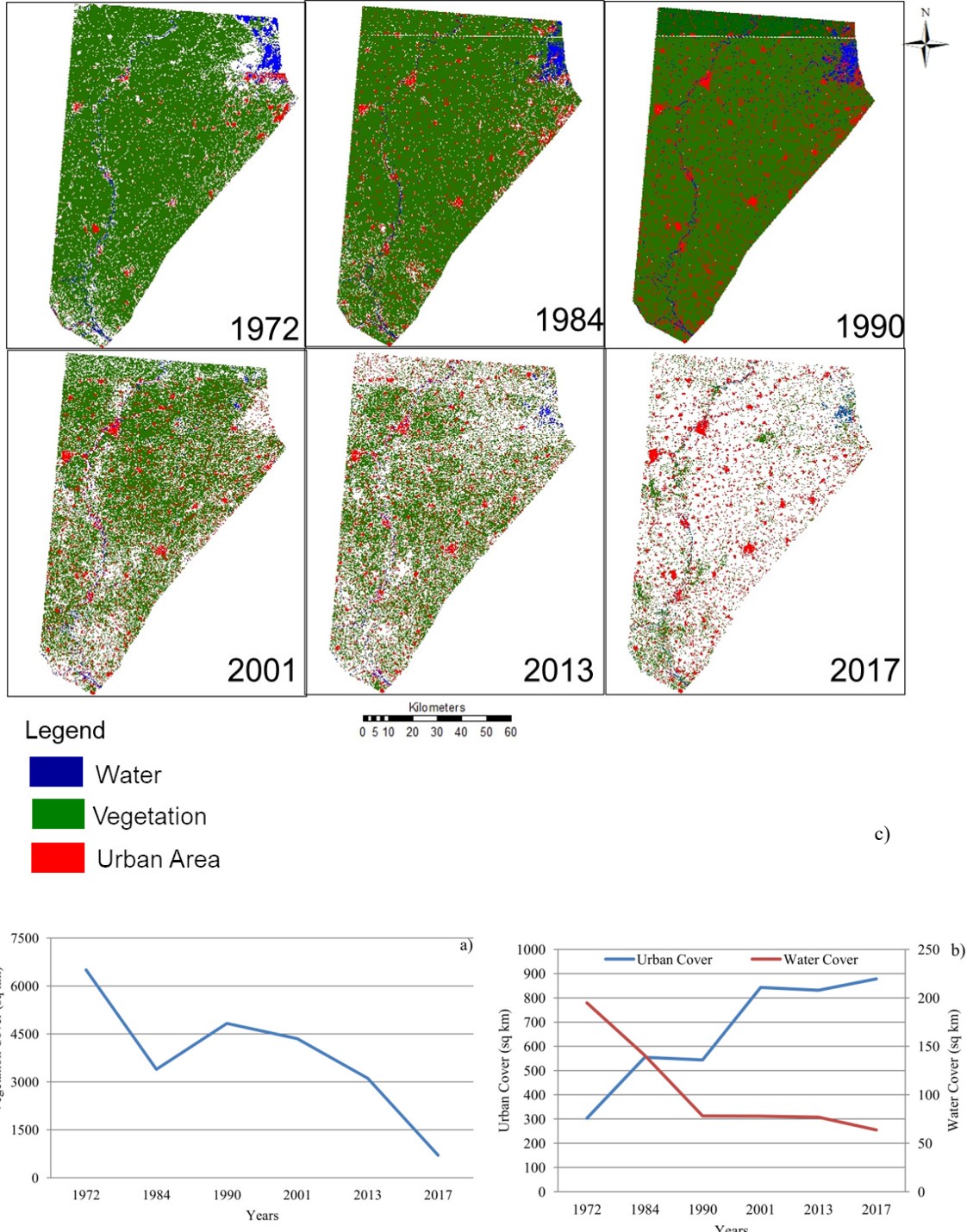

**Figure 7.** Multitemporal (1972–2017) analysis for the Damietta tributary of the Nile River illustrating spatial distribution of water, vegetation, and urban spread. Spatial statistics for (**a**) water, (**b**) vegetation, and (**c**) urban sprawl. Significant decline of vegetation cover and the shrinking of water spread is noted.

The population of the region saw a sharp rise from 1990 to 2017 (i.e., 60 million in the 1990s to >98 million in 2017). In the last 30 years, around 30 million people were added to the population records. Migration trends (international migration stock to the total population increased from 1990 to 2017, as shown in Figure 9) reflect the influence of social and political conflicts including the Syrian crisis as a contributing source. Most displaced people (including the migration flow from the conflict zones

such as Syria) are hosted in urban centers in the Nile delta region. Internal and international migration in the region is on the rise from 2005 and continues to escalate [49,57].

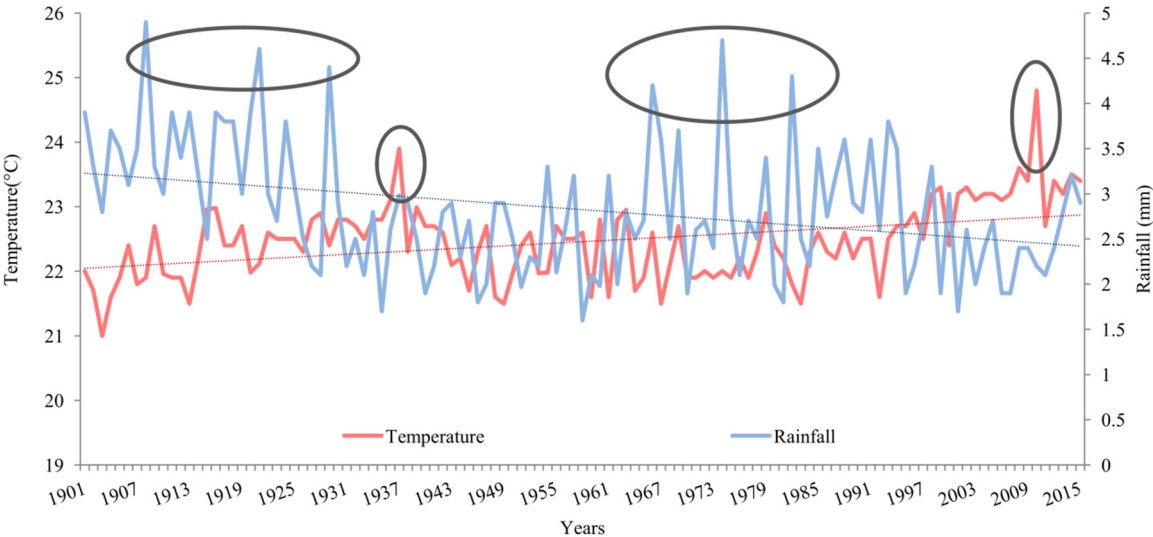

**Figure 8.** Climate variations (temperature and rainfall) trends in Egypt (data adapted from Climate Change Knowledge Portal). The dry spells in the region compounded with the influence of rising temperatures have impacted the agriculture and nature-based livelihood sectors.

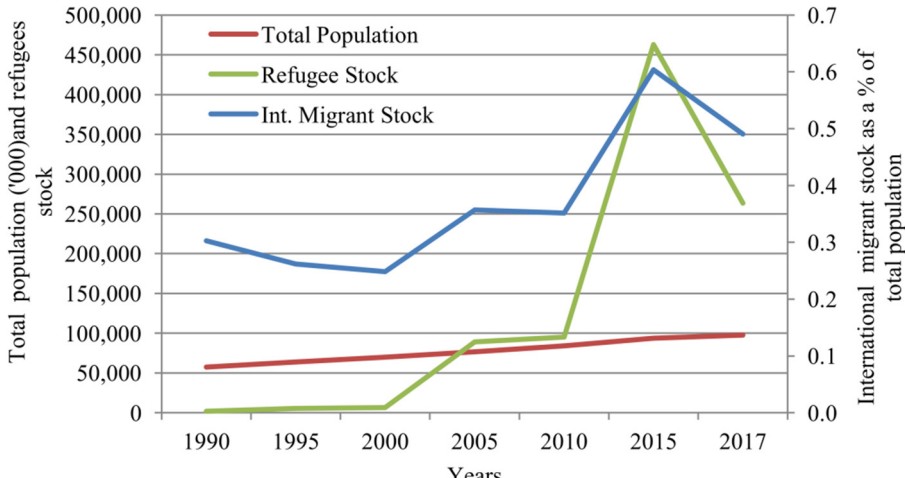

**Figure 9.** Trends in the population of the state (Egypt region) and the corresponding international migrants (including displaced people and refugees) from 1990–2017 (Data Source: International Organization for Migration (IOM) and United Nations Global Migration Database (UNGMD)).

Historically, migration in the region is driven by multiple factors. For example, the construction of the High Dam in Aswan (south of Egypt) caused the forced displacement of >40 Nubian villages as the area submerged and formed lakes spreading 350 km². Even now, more than 50 years later, the Nubian resettlement remains a highly controversial issue (more details in Table A1). Also, Iraqi invasion lead to resettlement of population in this region.n. Around 150,000 Iraqi resettlements in Egypt was noted between 2003 and 2005 (see the overall migration trend in the region in Figure 9). Reports show that the official numbers of Syrian refugees in Egypt to be about >100,000; however, the unofficial count in 2012 was >0.65 million. Most of the refugees are known to be hosted in the east and west of the city of Cairo in the delta region [58]. Water and climate crisis, combined with social and political dynamics, has directly and indirectly contributed to the flow of people and communities, according

to the argument by Gleick and colleagues. Additionally, Gleick et al. describe the link between the Syrian crisis and water insecurity [59]. Overall, this case study provides an opportunity to understand the dynamics of water and climate related crisis and conflicts and its influence on displacement of people and communities. Furthermore, how drivers/triggers, particularly, water-related causes and consequences and migration affecturbanization.

Case Study 3: The Aral Sea Region in Central Asia

The Aral Sea region is a typical case of myopic development planning. In 1976, the basin spread to 55,700 km$^2$. In the past three decades, the degeneration of the Aral Sea ecosystem, primarily the shrinking of the surface water spread (9830 km$^2$ in 2017), has been observed. These changes have split the Aral Sea in two, which includes the North (in Kazakhstan) and South Aral (in Uzbekistan). Adverse impacts of these hydrological changes are multifaceted, for example, many ports including the main port of Aralsk in the region were left dry and devoid of economic activity. The region has attracted a lot of scientific attention with some continuous monitoring efforts [60]. Nevertheless, a limited focus was given to the socio-economic consequences of this hydrological disaster.

Multi-temporal mapping for the region illustrates the spread of surface water and nearby ecosystems (Figure 10). The geospatial assessment shows water spread declined from 1973–2017. Furthermore, the mapping presents a visual layout of the hydrological dynamics. It is apparent that the hydrological flux will lead to various socio-economic consequences. For instance, in the south of the Aral Sea, the city of Mujnak reported an economic collapse when the Amu Darya flow was stalled in the 1960s. This event created a chronic water crisis in the urban center. Both rural and urban inhabitants, who relied on the Aral Sea for livelihood and income, suffered a loss of opportunity, causing a high unemployment rate, lack of access to freshwater, and forced migration [60]. This ecological disaster displaced >100,000 people and affected the health of >5 million people throughout the region [61]. Half of the population of the area was forced to migrate in search of better living conditions and employment [60]. Even today, the lack of potable water for inhabitants reflects the gradient of hydrological and socio-economic crises in the region.

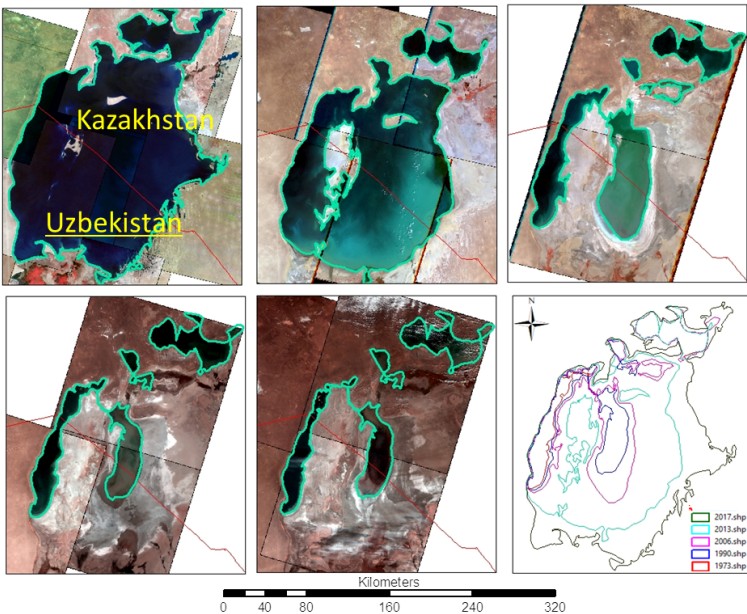

**Figure 10.** Geospatial analysis of the Aral Sea region (1973–2017) derived from the Landsat data reflecting the shrinking of surface water and partition of the lake's surface due to various natural and anthropogenic factors.

It is estimated that more than 25 percent of the population of the Aral Sea region has migrated as refugees outside the region or to locations in Kazakhstan or Uzbekistan [62]. Bulesheva and Joldasov reported displacement of 0.5 million people in 14 years, and situations wherein individuals and communities from Uzbekistan migrated to Kazakhstan [63]. Migration from rural to urban areas within the basin is significant as the supply of food and potable water and other provisioning facilities are confined in cities or urban centers. The rapid urbanization in the region is one outcome of the crisis stemming from ineffective water management interventions. Close to 50 percent of people in the region have shifted to urban areas, and the rural areas located close to the Aral Sea shoreline have been abandoned [63]. This case study presents an opportunity to comment on the rural-urban migration and how migration could drive rapid urbanization.

The population directly affected by the Aral Sea crisis is estimated around 5 million. To refill the Aral Sea to the base shoreline (recorded in the 1960s) will require 450 percent more water inflow compared to the current flow [64]. The existing economic growth strategies of Uzbekistan and Turkmenistan, the region that depends on the cotton production industry and one of the most demanding sectors for water utilization for economic growth, may find the commitments to the Aral regeneration agenda as challenging [65]. It will also be hard for other states surrounding the lake basin to negotiate new inflow dynamic thresholds, given similar or related economic growth plans. Water stress in the Aral Sea region can be viewed as the byproduct of the agricultural water requirement of cotton and other water-intensive crops [66]. To put these numbers into perspective, note that about 8000 liters of water are needed to produce 1 kg of the consumed product of cotton, which means that to create one cotton shirt (weight of 250 g), approximately 2000 liters of water is needed [67].

The water -migration nexus in this case reflects on the disruption of water availability and storage, lack of appropriate water provisioning for food production, land and water degradation issues, desertification, and water insecurity scenarios. It also reflects how these dimensions have acted as direct and indirect drivers to migration and other socioeconomic outcomes. For instance, the fishery sector severely impacted by loss of fish-based livelihoods and income due tothe increase in salinity, pesticide pollution draining from the surrounding agricultural land, and health concerns from the water pollution [60]. The multi-temporal assessment also reflected the impact on the surrounding ecosystems and native vegetation. The region observed a rise in halophytes (plant communities of saline land) and xerophytes (plants that grow with very little moisture) and the emergence of solonchak or salt pans (seen as white color patches in the digital images shown in Figure 10). Furthermore, the climate variability (i.e., the increasing temperature and precipitation flux) aggravates the effects of water crisis (Figure 12a–c), which is manifested as dry conditions or floods.

Since 1900, the population of the region has increased seven times (50 million). Furthermore, irrigated lands have also doubled (7.5–7.9 million hectares), and the available water quantity is compromised due to mismanagement [68]. A closer look at the United Nations Migration Statistics in 2006 helps to understand the regional migration trends. For instance, in Uzbekistan, the number of refugees (often stated as the victims of the Aral Sea crisis) increased from 1995 to 2009 [69]. In Kazakhstan, the absolute number of refugees was less (15000 people in 2000), but still a significant number when compared to the population of that country. Figure 11a,b shows the migration trends in the region including the details of refugees and asylum seekers. The ethnic population (Karakalpak, native to the area) is also influenced by consequences of the water crisis, for instance, through social (forced migration), economic (loss their traditional livelihood), and health-related outcomes [70]. The explanation of the drought of 2000–2001 by Ataniyazova points to the severe impact on health and economic aspects. Rice and cotton plantations were also severely impacted by dry conditions. Besides the degradation of the agricultural sector, the study documented that >50 percent of Amu Darya irrigation water was lost due to the lack of canal lining, excessive filtration, and evaporation [71]. Overall, this case study argues for the maps and visuals as a good foundation to create narratives to explain the spillover impacts that hydrological events have had on the socioeconomic outcomes, which include migration.

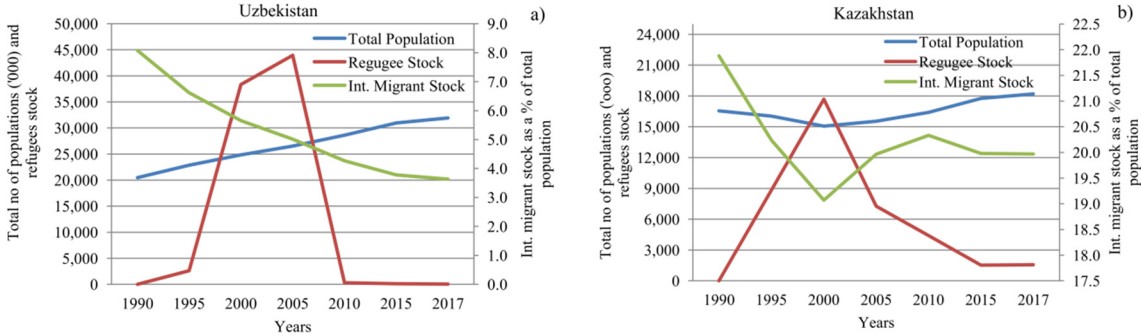

**Figure 11.** Trends of international migration and estimated refugee stock (including asylum seekers) in Kazakhstan (**a**) and Uzbekistan (**b**) from 1990 to 2017.

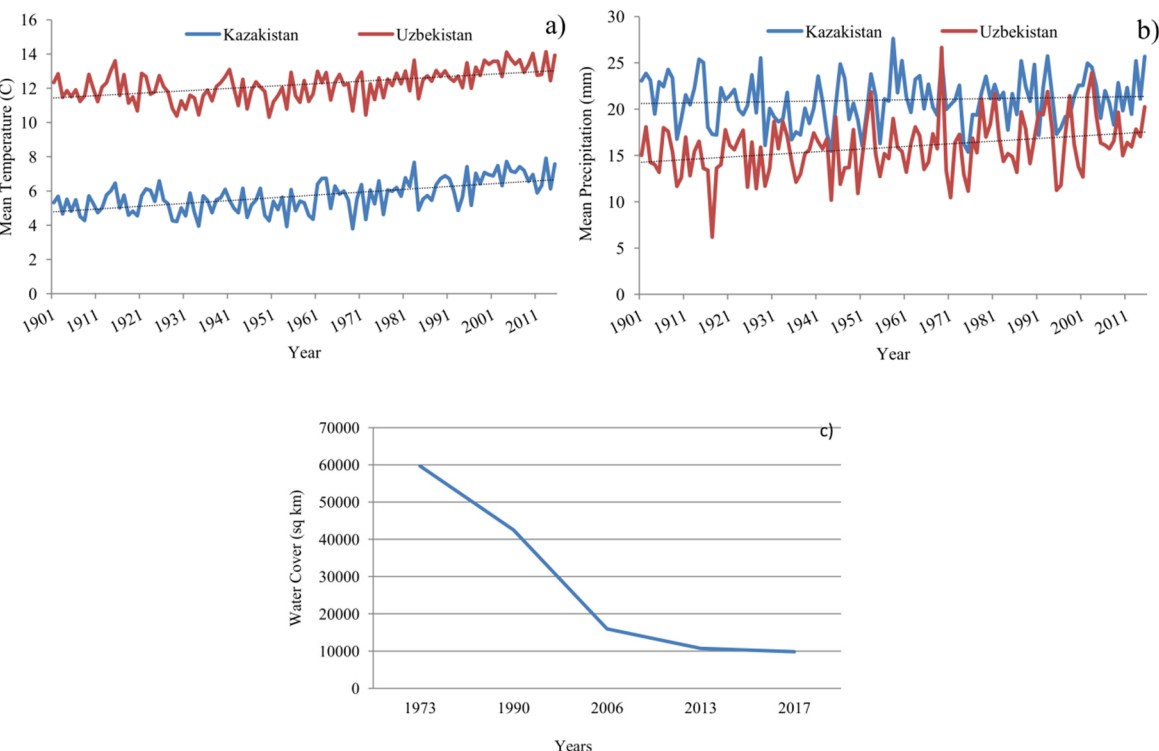

**Figure 12.** Climatic variability trends for Kazakhstan and Uzbekistan (1901–2015). (**a**) mean temperature, (**b**) precipitation. The average monthly temperature and rainfall (data sources: Climate Portal [72] and, (**c**) water cover spread (derived from remote sensing analysis/spatial statistics).

*4.2. Phase II: The Global Scale Analysis Based on Global Conflict Risk Index (GCRI)*

The geospatial representation of the GCRI provides a comprehensive global overview of settings that influence risk and conflicts in various social, environmental, and political settings (Figure 13). Selected parameters from the index illustrated in a GIS platform allows for the risk to be assessed and to compare conflicts activated by various dimensions (e.g., social, political, environmental). The indices embedded in the GCRI also illustrate the disputes arising from the shared water supplies. Notably, managing shared water systems remains a critical challenge and pertinent political issue worldwide, and is an issue known to trigger hostility among countries [73–75]. Additionally, SDGs and their related targets, mainly SDG 16 (aiming promotion of peaceful and inclusive societies for sustainable development and building effective, accountable, and inclusive institutions), are closely connected with the conflict and migration context in a relational manner. The high rate of conflict or diversity of conflicts can act as a barrier to the implementation of SDG 16. There can be many triggers to a

conflict. In the context of water, conflict can be triggered by the lack of availability of water or due to the deterioration of water quality supplied for drinking or even by unfair water pricing. Hence, such conflicts directly interface with the targets outlined in the SDG 6 portfolio. For instance, target 6b, which focuses on supporting and strengthening the participation of local communities in water management related decision making.

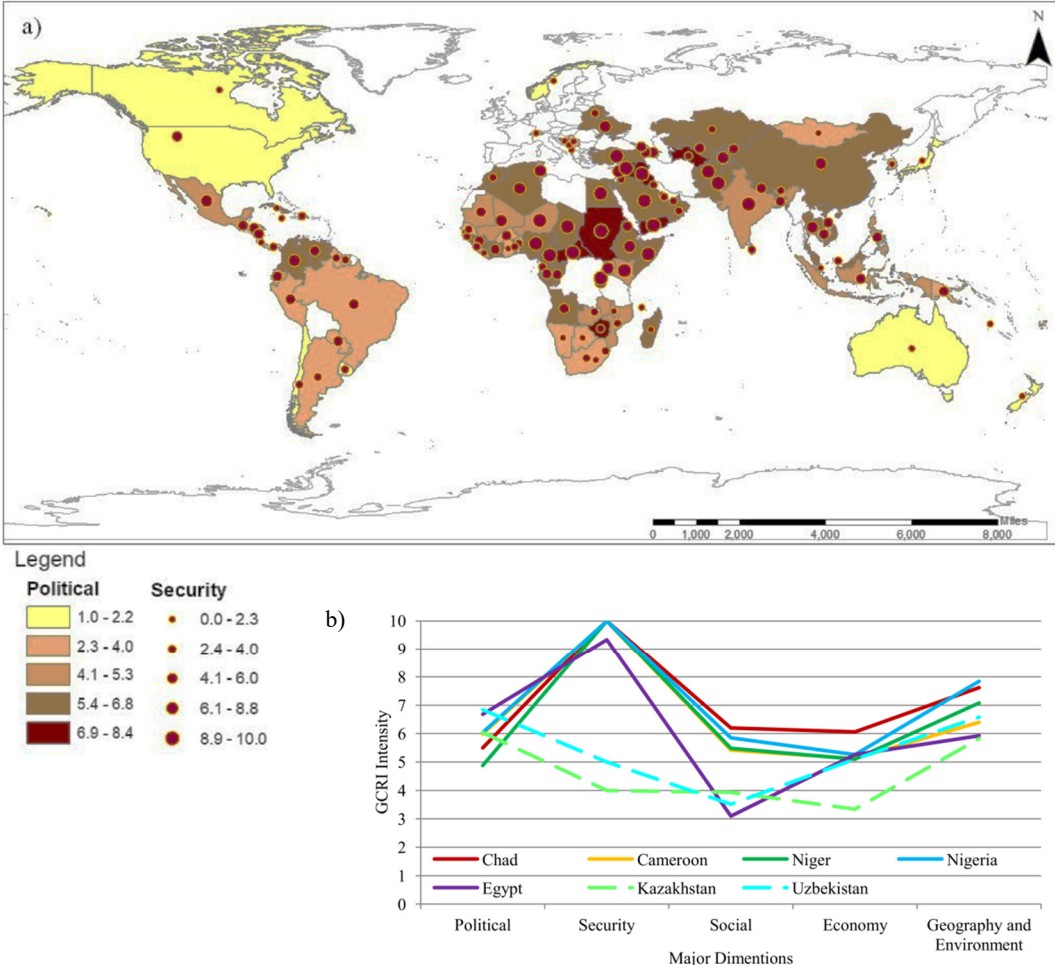

**Figure 13.** (**a**) Global Conflict Risk Index (GCRI) projected using a GIS platform reflecting on the political and security interlinkages (a low value and small dot equal more stability security and vice versa, the blank area is no data); (**b**) GCRI trends in selected areas (representing the case studies sites) of Africa and Asia.

In the above context, the selected input parameters (indicators reflecting on the social, economic, environmental, political, and security-related risks) of the GCRI could provide a comparative context of various threats to human development and security in multiple geographic, socio-economic, and socio-political settings. For instance, the ecological/environmental risk is a growing concern in most case study regions, namely Chad, Cameroon, Niger, Nigeria, Egypt, Kazakhstan, and Uzbekistan. These regions also experience and report high migration. Kazakhstan and Uzbekistan reported high values in the political, geography, and environmental categories of the index (Figure 13b). Thereby, GCRI provides an additional narrative to explain the ecological collapse and subsequent conflicts in the case of the Aral Sea (Case study 3) that led to dramatic social and economic consequences in a time continuum. The GCRI analysis fits well with the SDG 16 agenda, and shows potential to serve as a measure to populate the requirements set out in SDG 16 progress monitoring. For instance, SDG targets 16.1.2 (managing conflicts), 16.3 (building up-to-date information on drivers and promote

policies for decision support), 16.6, and 16.7 talk about the development of effective, accountable, and transparent institutions and ensuring inclusive, participatory, and representative decision-making at all levels (including developmental challenges such as migration, water, and climate crisis) [72].

## 5. Discussion

Water crisis scenarios such as water quality, challenges related to availability, and extreme water situations all trigger migration, both directly and indirectly. To this, an integrated assessment of various aspects such as the geographic location, hydrological functions, ecological issues, and socio-economic dynamics of water and migration becomes essential. This study employs both quantitative and visual dimensions (e.g., maps, spatial statistics) as evidence toward an enhanced understanding of the water-migration connections and outlines some key points as below:

(a) A geospatial platform, tools, and data allows integration of different data sources including remote sensing images over a long period, further allowing the integration of climate variables and other quantitative data such as GCRI index values, and thus enable the assessment of water-migration interlinkages (nexus);

(b) The GIS platform serves as an excellent tool to showcase interlinkages. The maps or visual material that illustrate the water crisis scenarios can act as a practical tool to equip policymakers with information to better manage water crisis-related impacts and outcomes; for which, one outcome may be migration;

(c) Water and migration, when analyzed within the both physical and human geography (biophysical characterization, socioeconomics, socio-political and socio-cultural) perspective, connects well with the SDG agenda. For example, the case studies demonstrate the movement of people and communities from rural to urban areas and thus, a straight connection with urbanization.

(d) Changing trends in water use (quantitative) due to migration can lead to competing and conflicting water use discourse. As such, water and migration-related decision making needs to be participatory and consultative, particularly in the cases related to the geopolitical crisis.

In relation to point (d) in the above list, let us examine the case of Jordan. From 1990 to 2008, Jordan's population grew by 86% and with the influx of people, whereby close to a million refugees contributed to placing pressure on the water resources that were already scarce, exploited, or not managed efficiently, raised global upheaval [76]. The case study analysis in phase 1 of this paper comments on the land–water dynamics using a geospatial exercise, a multitemporal analysis, to provide an improved understanding of the triggers to migration and to some extent, the impacts of migration. The objectives outlined for both phases also offer a description to support SDG targets (6.4, 6.5 11.5, 11.B, and 16.7) implementation planning. While doing so, the case studies reflect upon the diverse impact of anthropogenic interventions (human activity), water crisis, and climate change variability, and how these acts as triggers to migration. For instance, Lake Chad and the Aral Sea are classic examples of 'poor or unsustainable water management' and demonstrate a 'lack of coherence' in long-term natural (water management) resource planning. The lack of integrated development planning in land and water management often leads to impacts like inequality, injustices, and outcomes such as forced displacement of individuals and communities that have long inhabited the region. Furthermore, vulnerable populations with socio-economic dependence on water resources and aquatic ecosystems for livelihood and income generation are most at stake from pressures of water and climate crisis. In this context, the Integrated Water Resource Management for Zambia (IWAREMA) project presents an excellent example of an integrated assessment applied in the sector of wildlife, agricultural activities, fisheries, and tourism to support Zambia's Department of Water Affairs [77]. Such examples of an integrated framework for natural resource management are available for other states and communities to refer to and adapt, with GIS as a supporting tool.

The increase in irrigation demand, both in developed and developing economies, and lack of or irregularities in systematic regulation and enforcement mechanisms of water management, has led

to episodes of water crisis. Additionally, the impacts of climate variability, which include declining trends in precipitation and increasing trends of temperature, exacerbate water-crisis, and risk exposure for people and communities. During the dry season, the water supply–demand dynamics are more complicated due to limited water availability. The geospatial analysis allows for an assessment of seasonal dynamics, thereby to understand how water demand and supply varies, noting that these variations can influence socioeconomic scenarios such as conflicts and migration [17]. For instance, the Aral Sea case study illustrates how short-tenured agricultural expansion policies targeting fast economic growth (shift to non-food crops) have led to food and water insecurity as well as a loss of livelihood and income and people moving out of their homelands. In another instance, the countries around Lake Chad and the Nile River are facing high security-related risks such as internal conflicts, violent conflicts, political turmoil, and social tensions due to a variety of reasons. The case studies selected for this paper have attracted a lot of international attention, and global development agencies worldwide are supporting interventions for their stability and sustainability. The Hague Declaration is an initiative promoting the goal of coordinating migration, climate change, and urban resilience as well as supporting joint risk assessment in Lake Chad [78].

The global analysis in phase 2 of the analysis was useful to provide a comparative context, for example, a high GCRI score for the case study regions (except Kazakhstan, Uzbekistan) shows that their exposure and vulnerability to conflicts is high. The rating also provides an indication (serving as proxy) to the political stability, security, social stability, economic, and environmental stability (or instability/conflict) context. The GCRI evaluation is relevant since many parameters of this index relate to the settings that apply to migration as a direct or indirect outcome. For instance, in the case of Lake Chad, political crisis is mainly triggered by civil conflicts such as the Boko Haram insurgency. Such events, in turn, acting as indirect drivers, can intensify the impact of land and water crisis scenarios and, subsequently, migration [79]. Two key migration trends identified in the Lake Chad basin were internal displacement, and international migration flows, whereby an estimated 2.5 million people have been displaced in Cameroon, Chad, Niger, and Nigeria as of insurgency (>90 % internal migration). Migration across national borders remains restricted in comparison to internal displacement. The movement of people along the borderline of the lake with Nigeria is reported [79].

Migration related decision-making is not a linear relationship with one driving factor. Instead, several direct and indirect drivers act in tandem. The WWDR (2019) shows how migration, triggered as the result of political conflicts, can aggravate water, food, and climate crises and, thus, the health, well-being, and stability of populations and communities [9]. For instance, displaced people live in conditions favourable for spreading infectious diseases such as cholera, measles, meningitis, yellow fever, and water-related diseases such as malaria, yellow fever, dengue fever, etc. [80]. Having access to up-to-date information to understand the complexity of situations and quantitative assessment can allow decision-makers to study the interlinkages between migration and transitions such as the expansion of urban areas or a loss of forest and agricultural regions. Such information can also enable a better analysis of the risks caused by issues related to water availability and flux in surface water, flooding, and climate extremes [77].

Furthermore, geospatial indicators are explained as a surrogate to assess and monitor the water-related SDG agenda: SDG 6.1 and SDG 6.2 talk about universal and equitable access to safe and affordable drinking water and adequate and equitable sanitation and hygiene; SDG 6.4.1 (water-use efficiency across all sectors and for ensuring the supply of freshwater to address water scarcity and to reduce the number of people suffering from water scarcity substantially), 6.4.2 (water stress), and 6.5.1 (Integrated Water Resources Management). SDG 6.3 focuses on the calibration of water bodies with good ambient water quality. Thus, remote sensing sensors can help to monitor surface water quality, providing a cost-effective monitoring solution [81].

The SDG agenda, with the framing of 'for all,' emphasizes the needs of vulnerable communities and displaced populations (migrants). In this context, both phases of this study echo the potential of applying a geospatial framework toward an improved understanding of water-related migration,

and to support the implementation of SDG goals and targets. More so, SDG 11a (positive economic, social and environmental links between urban, peri-urban, and rural areas through strengthening national and regional development planning) and SDG 11.b (increasing the number of cities and human settlements adopting and implementing integrated policies and plans toward inclusion, resource efficiency, mitigation, and adaptation to climate change and resilience inhabitations). SDG 11.5 (to reduce people affected and substantially decrease the direct economic losses caused by disasters, including water-crisis events) also points to the water and climate crisis-triggered migration context. Reckoning that forced migration can push people to live in slums, informal settlements, or inadequate housing facilities, the migration dimension is clear. SDG 11.3, on inclusive and sustainable urbanization and capacity for participatory, integrated, and sustainable human settlement planning and management, is also relevant in the case of migration, particularly the flow of people from rural to urban areas. Therefore, the investigation of water-migration interlinkages (nexus) and how that influences urban growth can contribute to smart and inclusive urban planning.

The case study regions of Lake Chad and the Aral Sea region are arid, as such they require water management strategies to consider the wide-ranging consequences of the water crisis, of which migration is among the other key impacts. In the case of the Nile River, the discussion of the 'Aswan Dam' points to political conflict in the transboundary basin. Water allocation and sharing mechanisms in the basin states have a controversial and politically sensitive context. The necessity of a balance for managing water needs to various sectors, communities, and countries, irrespective of their biophysical advantage (upstream-downstream conundrum), also applies. Water-related interventions such as the alteration of watercourses for developing irrigation systems, redistribution of water allotments among sharing countries, and dam construction are a few examples that explain the high GRCI values, especially in the categories of environmental and security indicators [82]. A global overview of the GCRI is well-positioned as support for SDG target 16.8 (strengthening the participation of developing countries in the institutions of global governance) by providing knowledgeable arguments explaining water-migration interlinkages. Additionally, the plan for SDG 16.10 and 16.b (widening the scope of international agreements and promotion of laws and policies for sustainable development) can benefit from quantitative information and visual representation to reflect upon these interlinkages.

## 6. Conclusions

Migration is an international developmental challenge noted in both the states of origin (mostly developing countries) and hosting sites/states. However, the current literature on migration captures the water-migration nexus to a limited extent. In this context, this study provided a novel approach to examine the water-migration scenarios by applying a geospatial approach. The two-phased method presented in this paper can help toward an enhanced understanding of the interplay between the direct and indirect drivers of water and climate crisis-driven migration at different levels, local to global. The multitemporal analysis applied to examine the case studies helps to depict the new realities of migration in various geographic, socio-economic, and socio-political settings. The narratives outlined in this paper, along with the case studies, aim to describe a set of evidence to discuss possible solutions for mitigation, adaptation, and resilience planning. Water crisis in each of the case studies analyzed over a long period showed how anthropogenic or natural drivers exacerbated by climate change deepen the gravity of impact on people and populations that, in turn, affect their decision to migrate. Although it remains challenging to pinpoint the primary driver of migration in water crisis- scenarios, an understanding of the interlinkages within various social, geographic, economic, and development realities remain significant for migration assessment. In this context, the GCRI index reflects a global trend of environmental and climate-related conflicts. The value of the GCRI index is noted as high for selected regions in Africa and Asia. In most instances, a high conflict region corresponds to high migration flow. Furthermore, the study demonstrates that the SDG 6, SDG 11, and SDG 16 targets bear specific orientation with water and migration scenarios.

This study reiterates the application of geospatial platform, multitemporal analysis, and use of proxy spatial indicators to assess the water crisis scenarios. The geospatial indicators such as the spatial extent of water and aquatic vegetation serve as a proxy to understand the water–land-human interaction in various socio-ecological (Chad, Aral, and the Nile delta) situations. The case studies flag the need for proper and comprehensive migration assessments where geospatial data can provide a cost-effective way to generate evidence and help design 'solution-oriented' support systems. Additionally, the proposed frameworks (case study approach + proxy indicators analyzed using geospatial data and tools) for migration assessments can be useful to transform empirical and synthesis work into tangible evidence (maps, trends, etc.). Such a foundation, in turn, can assist in the creation of up-to-date data, integrated research assignments, and inclusive policy outcomes for migration-related challenges. The geospatial analysis also showed urban sprawl growth, which aligns with the rapid urbanization concerns voiced by global leaders and development agencies. Overall, this paper provides a scientific scholarship to the global challenge of migration.

**Author Contributions:** Conceptualization, N.N.; methodology, R.B.; software, R.B.; validation, N.N., R.B.; formal analysis, R.B.; investigation N.N. and R.B.; resources, N.N and R.B.; data curation, N.N. and R.B.; writing—original draft preparation, N.N. and R.B.; writing—review and editing, N.N.; visualization, R.B.; supervision, N.N.; work administration, N.N.; funding acquisition, UNU INWEH. All authors have read and approved the final manuscript.

**Funding:** UNU INWEH would like to acknowledge the funding and support of Global Affairs Canada to its research, policy, and development mandate, programs, and projects.

**Acknowledgments:** We wish to thank all researchers at UNU INWEH who have contributed to the paper through discussions and review. We would also like to acknowledge the contribution of McMaster University scholars, Danielle Liao, Sami Kurani, and Chloe Wale. Thanks also to Panthea Pouramin and Pallavi for feedback on the paper.

**Conflicts of Interest:** The authors declare no conflict of interest. The funders had no role in the design of the study; in the collection, analyses, or interpretation of data; in the writing of the manuscript, or in the decision to publish the results.

## Appendix A

**Table A1.** Selected events of human mobility in the Nile region, the most conspicuous of which is the Nubian resettlement, displacement, and insecurity saga of >50 years (information adapted from Serag, 2016).

| Year | Event and Impact | Water Connect |
|------|------------------|---------------|
| 1950s | During the Suez War in 1956, Port Said—one of the main battlegrounds—severely damaged during the event—no official decision for eviction formed—many families were displaced from Egypt internally, mainly from the Delta [58] | Loss of livelihood and income generation activities from the water. |
| 1960s | In the 1960s, 55,000 Nubians were internally displaced in 1964 due to the construction of the High Dam in Aswan in the south of Egypt, where a42 Nubian villages were submerged by the formation of a 350 km$^2$ lake area [58] | Linked with water quantity management and the possibility of extreme water scenarios, this was reported as the case of classified as forced displacement |
| 2000–2005 | Invasion in Iraq and the spread of sectarian violence resulted in nearly 120,000 to 130,000 Iraqis resetting in Egypt [58] | Some reports quote poisoning of water bodies as a strategy applied in the violent conflicts |
| 2010–2017 | In 2012, the official numbers of Syrian refugees in Egypt counted for about 115,000 to 120,000. Unofficial numbers >650,000. Hotspots: east and west of the city of Cairo in the delta region [58] | Conflicts in the Syria region have some deep roots in water scarcity and persistent droughts |

**Table A2.** Details of the Landsat data used in the study.

| Lake Chad | | Aral Sea | | Nile River | |
|---|---|---|---|---|---|
| Path-Row | Dates | Path-Row | Dates | Path-Row | Dates |
| 199-50 | 10-Oct-73 | 172-29 | 1-Oct-73 | 190-38 | 31-Aug-72 |
| 199-51 | 10-Oct-73 | 174-30 | 3-Oct-73 | 190-39 | 4-Jan-73 |
| 199-51 | 10-Oct-73 | 173-30 | 14-Sep-73 | 176-38 | 20-Sep-84 |
| 198-51 | 30-Apr-73 | 174-29 | 15-Sep-73 | 176-39 | 2-Jul-84 |
| 198-50 | 19-Dec-75 | 173-28 | 29-May-73 | 176-39 | 20-Sep-84 |
| 198-51 | 8-Oct-75 | 173-27 | 10-May-75 | 176-39 | 24-Jan-90 |
| 199-50 | 9-Oct-75 | 173-28 | 10-May-75 | 176-38 | 25-Feb-90 |
| 198-50 | 8-Oct-76 | 173-29 | 10-May-75 | 161-28 | 24-Sep-06 |
| 198-52 | 19-Nov-79 | 173-30 | 10-May-75 | 176-38 | 14-Jan-01 |
| 199-50 | 20-Nov-79 | 174-27 | 11-May-75 | 177-39 | 23-Dec-01 |
| 199-51 | 20-Nov-79 | 175-28 | 12-May-75 | 176-39 | 14-Jan-01 |
| 200-50 | 23-Mar-79 | 175-29 | 12-May-75 | 162-28 | 4-Oct-13 |
| 184-51 | 20-Oct-86 | 173-30 | 15-Jun-75 | 161-28 | 6-May-13 |
| 185-50 | 24-Aug-86 | 160-29 | 4-Aug-90 | 191-29 | 6-May-13 |
| 186-50 | 2-Oct-86 | 161-29 | 12-Sep-90 | 162-29 | 17-Aug-13 |
| 186-51 | 2-Oct-86 | 160-28 | 19-Jul-90 | 161-28 | 24-Oct-17 |
| 185-51 | 7-Nov-87 | 162-28 | 21-Oct-90 | 161-29 | 24-Oct-17 |
| 185-51 | 31-Dec-01 | 162-29 | 21-Oct-90 | 162-28 | 6-Apr-17 |
| 184-51 | 6-Nov-01 | 161-28 | 27-Aug-90 | | |
| 186-51 | 17-Sep-01 | 161-29 | 27-Aug-90 | | |
| 185-50 | 21-May-01 | 162-28 | 30-May-90 | | |
| 186-50 | 12-May-01 | 162-29 | 30-May-90 | | |
| 186-50 | 21-May-13 | 162-28 | 7-Sep-06 | | |
| 186-50 | 25-Aug-13 | 162-29 | 7-Sep-06 | | |
| 186-50 | 31-Oct-14 | 160-28 | 24-Aug-06 | | |
| 185-50 | 1-Jan-17 | 160-29 | 24-Aug-06 | | |
| 184-50 | 16-Apr-17 | 161-28 | 31-Aug-06 | | |
| 186-50 | 17-Aug-16 | 160-28 | 13-Apr-13 | | |
| 184-50 | 18-May-17 | 162-28 | 6-Apr-13 | | |
| 185-51 | 1-Jan-17 | 162-29 | 6-Apr-13 | | |
| 184-51 | 21-Jul-17 | 161-28 | 20-Apr-13 | | |
| 185-51 | 22-May-16 | 184-51 | 23-Nov-16 | | |
| 186-50 | 29-Mar-17 | 186-50 | 5-Nov-16 | | |
| 184-51 | 2-May-17 | 184-50 | 6-Aug-17 | | |
| | | 184-51 | 6-Aug-17 | | |
| | | 186-50 | 30-Apr-17 | | |

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
