# Peer review of "Geospatial Assessment of Water-Migration Scenarios in the Context of Sustainable Development Goals (SDGs) 6, 11, and 16"

_remotesensing, doi:10.3390/rs12091376_

Round 1
Reviewer 1 Report
I would like to thank you for your submission. While I consider your paper timely, significant and innovative in terms of the addressed Sustainable Development Goals and regions of interest, I found it to be problematic in terms of written (text) and visual (Figures) representations. The latter certainly decreases its real merit and value. While the Introduction and Discussion-Conclusions parts are generally well written, parts 2., 3., and 4. lack clear thought, good English and hence decrease broad understanding of the concepts that you want to layout. Also, as a last point, you include several discussions, assumptions and speculations in the Results section which belong to the Discussion part. I have included my suggestions/recommendations/corrections/issues as highlights along with related text boxes in the attached pdf. All in all, I would like to point your focus especially on improving the Methods and Results sections, along with addressing the highlighted text in all sections, for your submission to be appropriate for publication in the Remote Sensing journal.

Author Response
I want to thank you for your submission. While I consider your paper timely, significant and innovative in terms of the addressed Sustainable Development Goals and regions of interest, I found it to be problematic in terms of written (text) and visual (Figures) representations.
Response: The text section and figure are revised as per the suggestions provided in the pdf version. Almost all comments and feedback highlighted in the review (pdf document) were considered appropriately and are reflected in the revised version
While the Introduction and Discussion-Conclusions parts are generally well written, parts 2., 3., and 4. lack clear thought, good English and hence decrease broad understanding of the concepts that you want to layout.
Response: Thank you for the feedback. As suggested these parts were rewritten and organised to ensure better structure, flow and clarity, for example parts of the text from the result section moved to the discussion.
Also, as a last point, you include several discussions, assumptions and speculations in the Results section which belong to the Discussion part. I have included my suggestions/recommendations/corrections/issues as highlights along with related text boxes in the attached pdf. All in all, I would like to point your focus especially on improving the Methods and Results sections.
Response: All detailed comments highlighted in the pdf documents were duly considered during the revision, the results and the discussion section is reorganized to fit the review comments and all suggested edits and changes marked in the pdf file is accounted to produce the revised document.
Addressing the highlighted text in all sections, for your submission to be appropriate for publication in the Remote Sensing journal.
Response: We would like to thank the reviewer for all the constructive comments that has helped to fine-tune the paper, all comments were carefully considered during the revision process.
Reviewer 2 Report
In Chp. 3. Data and Methods, pag. 5, rows 202-204: please provide some information about the acquisitions time period of the satellite imageries.
In Chp. 3. Data and Methods, pag. 5, row 205: please explain more in detail the parameter “clarity”.
In Chp. 3. Data and Methods, pag. 5, rows 224: “In this study,> 0 threshold value has been used to delineate the surface water area”; please provide more specific information for the positive threshold value assigned to delineate the water areas in different environments.
In chp. 4. Results, subchp. 4.1. The Lake Chad, pag. 7: the title of this subchp. does not adequately reflect the content, taking into account that in this part the authors describe also the results of the Nile Delta Region and the The Aral Sea region; please change the subchp 4.1. title.
In chp. 4. Results, subchp. 4.1. The Lake Chad, pag. 8, rows 297-298: „The annual mean temperature, precipitation changes and population trends are (Figure 4) also align with the interpretation of geospatial trends.”; please support more in details this afirmation.
In chp. 4. Results, subchp. 4.1. The Lake Chad, pag. 8; the figure 3-b is not cited in the manuscript text.
In chp. 4. Results, subchp. 4.1. The Lake Chad, pag.8, rows 312 – 314: The sentence „The spatial distribution of population around the lake derived using Landsat (2017), i.e. by buffering 100 km around lake boundary and Google Earth data, illustrated in (Figure 5), the habitation spread.” is unclear; please reprase it.
In chp. 4. Results, subchp. 4.1. The Lake Chad, pag. 10, row 346; the Figure 7b is not cited in the manuscript text.
In chp. 4. Results, subchp. 4.1. The Lake Chad, pag. 10, rows 368-370: the sentence „States The social and political conflicts, including the Syrian crisis, in the area often referred to as the contributing source of migration pathways.” is unclear; please refrase it.
In chp. 4. Results, subchp. 4.1. The Lake Chad,pag. 11, rows 379-380: the title of Figure 8 „ Climate variability and human migration dynamics in the Egypt region (data adapted from Climate Change Knowledge Portal and UN Migration Statistics database)”, does not adequately reflect the content of the figure. Please change the figure title.
In chp. 4. Results, subchp. 4.1. The Lake Chad,pag. 12, row 407: the sentence „In the 1970’s the basin spread was (55,700 km2 in 1976), that” is unclear; please refrase it.
In chp. 4. Results, subchp. 4.1. The Lake Chad,pag. 12, rows 413 -415: the sentence „The time series visual digitalization (Figure 9) adopted in our study reflects on the trends of land degradation (loss of vegetation) as well as some recovery efforts (measured as change in water spread in the north part of the Aral Sea)” is unclear ; why the figure 9 is cited here ?; please modify it.
In chp. 4. Results, subchp. 4.1. The Lake Chad,pag. 14, rows 448-449: Figure 10 is unclear. The five processed satellite images have to be associated with the analysed years (e.g.: a-1973, b-1990, c-2006, d-2013 and d-2017).
In chp. 4. Results, subchp. 4.1. The Lake Chad, pag. 14, rows 453-454: the sentence „The climate change impacts exacerbate the severe water crisis created by the shrinking of the lake- so do the increasing temperature trends (Figure 11).”. The trends of the precipitation evolution in figure 11 b) seem to be slightly increasing, at least for the period 1981 to 2011; please explain more in detail this behavior in relation to the shrinking of the Aral sea surface.
In chp. 4. Results, subchp. 4.1. The Lake Chad,pag. 14, rows 509: Figure 13. Please assign a) and b) to the two figures.
In chp. 6. Concluding Notes, pag. 19, rows 629-631: the sentence ”GCRI index re-projected in a geospatial domain supports to identify patterns of environmental and climate related conflicts, for selected regions in Africa and Asia – the value of the index is high” is unclear, please refrase it.
Author Response
Data and Methods, pag. 5, rows 202-204: please provide some information about the acquisitions time period of the satellite imageries.
Response: The suggestion is accounted for in the revised version and detailed in Annexure 2
In Chp. 3. Data and Methods, pag. 5, row 205: please explain more in detail the parameter “clarity”.
Response: Taking note of the suggestion, the term is explained in the revised version- visual clarity such as cloud-free scenes,
In Chp. 3. Data and Methods, pag. 5, rows 224: “In this study,> 0 threshold value has been used to delineate the surface water area”; please provide more specific information for the positive threshold value assigned to delineate the water areas in different environments.
Response: The section is modified to accommodate the suggestions.
In chp. 4. Results, subchp. 4.1. The Lake Chad, pag. 7: the title of this subchp. does not adequately reflect the content, taking into account that in this part the authors describe also the results of the Nile Delta Region and the The Aral Sea region; please change the subchp 4.1. title.
Response: Taking note of the suggestion, the title is changed in the revised version and the subsequent case studies are labelled accordingly
In chp. 4. Results, subchp. 4.1. The Lake Chad, pag. 8, rows 297-298: „The annual mean temperature, precipitation changes and population trends are (Figure 4) also align with the interpretation of geospatial trends.”; please support more in details this affirmation.
Response: Brief notes are added to explain this part
In chp. 4. Results, subchp. 4.1. The Lake Chad, pag. 8; the figure 3-b is not cited in the manuscript text.
Response: Added
In chp. 4. Results, subchp. 4.1. The Lake Chad, pag.8, rows 312 – 314: The sentence „The spatial distribution of population around the lake derived using Landsat (2017), i.e. by buffering 100 km around lake boundary and Google Earth data, illustrated in (Figure 5), the habitation spread.” is unclear; please reprase it.
Response: Rephrased
In chp. 4. Results, subchp. 4.1. The Lake Chad, pag. 10, row 346; the Figure 7b is not cited in the manuscript text.
Response: Added
In chp. 4. Results, subchp. 4.1. The Lake Chad, pag. 10, rows 368-370: the sentence „States The social and political conflicts, including the Syrian crisis, in the area often referred to as the contributing source of migration pathways.” is unclear; please refrase it.
Response: The suggestion is accounted for in the revised version- the sentence is rephrased
In chp. 4. Results, subchp. 4.1. The Lake Chad,pag. 11, rows 379-380: the title of Figure 8 „ Climate variability and human migration dynamics in the Egypt region (data adapted from Climate Change Knowledge Portal and UN Migration Statistics database)”, does not adequately reflect the content of the figure. Please change the figure title.
Response: The suggestion is accounted for in the revised version- the Figure is provided with a new title
In chp. 4. Results, subchp. 4.1. The Lake Chad,pag. 12, row 407: the sentence „In the 1970’s the basin spread was (55,700 km2 in 1976), that” is unclear; please refrase it.
Response: The suggestion is accounted for in the revised version- the sentence is rephrased
In chp. 4. Results, subchp. 4.1. The Lake Chad,pag. 12, rows 413 -415: the sentence „The time series visual digitalization (Figure 9) adopted in our study reflects on the trends of land degradation (loss of vegetation) as well as some recovery efforts (measured as change in water spread in the north part of the Aral Sea)” is unclear ; why the figure 9 is cited here ?; please modify it.
Response: The suggestion is accounted for in the revised version
In chp. 4. Results, subchp. 4.1. The Lake Chad,pag. 14, rows 448-449: Figure 10 is unclear. The five processed satellite images have to be associated with the analysed years (e.g.: a-1973, b-1990, c-2006, d-2013 and d-2017).
Response: The suggestion is accounted for in the revised version
In chp. 4. Results, subchp. 4.1. The Lake Chad, pag. 14, rows 453-454: the sentence „The climate change impacts exacerbate the severe water crisis created by the shrinking of the lake- so do the increasing temperature trends (Figure 11).”. The trends of the precipitation evolution in figure 11 b) seem to be slightly increasing, at least for the period 1981 to 2011; please explain more in detail this behavior in relation to the shrinking of the Aral sea surface.
Response: Figure 11 title is revised and explained provided on the point highlighted by the reviewer
In chp. 4. Results, subchp. 4.1. The Lake Chad,pag. 14, rows 509: Figure 13. Please assign a) and b) to the two figures.
Response: As per the suggestion, the Figure is labelled as ‘a’ and ‘b’
In chp. 6. Concluding Notes, pag. 19, rows 629-631: the sentence” GCRI index re-projected in a geospatial domain supports to identify patterns of environmental and climate related conflicts, for selected regions in Africa and Asia – the value of the index is high” is unclear, please rephrase it.
Response: As per the suggestion, the rows 629-631- are rephased in the revised version
Reviewer 3 Report
The article presents a study that relates migration to the availability of water in a multitemporal environment and its decision support system related to Sustainable Development Goals.
Although satellite images are used in this study and several index calculations are performed (Normalized differential vegetation, Normalized Difference Water, normalized built-up) it does not contribute anything new within the scope of the journal.
This is a GIS study in which different data sources are integrated, including some remote sensing to show the evolution of the surface of lakes or basic wetlands for human settlements. It is an article that justifies the analyzes within the framework of the objectives of sustainable development, since human geography perspective.
The article talks about time series of images in the wrong way. It is a multitemporal analysis but does not meet the premise of a time series that the images are equidistant from time, there must be a period and it is not met in any of the cases (1973, 1975, 1986, 2001, 2013, 2017) or (1972, 1984, 1990, 2001, 2013, 2107).
When describing the media, a set of images is mentioned, but the precise dates and how many of them are to be joined together, forming a mosaic to analyze the areas of interest. This information is quite important in relation with time-series analysis.
The form of citation in the document is not the one that defines the editorial, numerical citations are mixed with APA citations.
The article fits better in other journals such as IJGI also from MDPI or other geography journals.
The study is very ambitious with three use cases, but none of them develop properly in deep. It is also intended to address a dashboard for decision-making based on available public information that does not link correctly with the previous cases.
For all the above I propose that the authors review the document, adjust objectives and refer to another journal.
Author Response
The article presents a study that relates migration to the availability of water in a multitemporal environment and its decision support system related to Sustainable Development Goals.
Although satellite images are used in this study and several index calculations are performed (Normalized differential vegetation, Normalized Difference Water, normalized built-up) it does not contribute anything new within the scope of the journal.
Response: We note this point, the intent of the study is to illustrate how remote sensing and GIS data, tools and technologies can be applied for human migration assessment, particularly for the instances where migration is driven by land, water or climate crisis and to showcase spatial indicators/ processes that directly or indirectly ( NDVI, Water Index ) etc. serves as a reference for designing future migration studies. Overall, the study is new in the context of investigating the land-water-migration nexus using set of geospatial data, tools and technologies, wherein the tools and technologies may not be new in itself- however, the thematic application of these for migration studies had been
This is a GIS study in which different data sources are integrated, including some remote sensing to show the evolution of the surface of lakes or basic wetlands for human settlements. It is an article that justifies the analyzes within the framework of the objectives of sustainable development, since human geography perspective.
Response: The study is an application exercise that builds on freely available data and east, smart and scalable techniques that can adopted and applied by researchers from various disciplines, mainly migration studies.
The article talks about time series of images in the wrong way. It is a multitemporal analysis but does not meet the premise of a time series that the images are equidistant from time, there must be a period and it is not met in any of the cases (1973, 1975, 1986, 2001, 2013, 2017) or (1972, 1984, 1990, 2001, 2013, 2107).
Response: We do note this point, and can adjust the context to multitemporal analysis, the selection of years (time sequences) is based on data availability and key event related to the case studies (decadal analysis projected through a selected point in time). We think this suggestion is fine, but not a deciding one in designing and conduction geospatial research analysis.
When describing the media, the set of images is mentioned, but the precise dates and how many of them are to be joined together, forming a mosaic to analyze the areas of interest. This information is quite important in relation with time-series analysis.
Response: This is a good point and the details related to the same is now listed in Annexure 2 and related text added in the data and method section
The form of citation in the document is not the one that defines the editorial, numerical citations are mixed with APA citations.
Response: All citations are now formatted as per the MDPI guidelines
The article fits better in other journals such as IJGI also from MDPI or other geography journals.
Response: We have submitted our interest in consideration in the SDG focused special volume of Remote Sensing
The study is very ambitious with three use cases, but none of them develop properly in deep. It is also intended to address a dashboard for decision-making based on available public information that does not link correctly with the previous cases.
Response: Parts of the script reflecting the connection are rewritten to explain the missing links, the case study approach is method applied to reflect on the diversity of migration scenarios in different geographic settings, and to revisit these known cases for assessing the migration context.
For all the above I propose that the authors review the document, adjust objectives and refer to another journal.
Response: The new version of the script is revised with assistance from a native speaker and the text/content adjusted for clarifying the points raised in the review. We find the script suitable for the SDG special volume.
Round 2
Reviewer 1 Report
Dear authors,
Thank you for your amendments and responses. Despite your assurance that you have appropriately considered the majority of my comments and that you have accordingly rewritten and logically structured your draft, I still hold major concerns for the latter. I still feel that you are missing a logical structure and a good natural and scientific cohesion and legitimacy between the individual parts, despite the first round of reviews and revisions. My greatest issue arises from the carelessness with which I feel you have approached a large part of your draft, which does not realistically reflect the level of published papers in the Remote Sensing journal. I would suggest here to address more attentively and thoroughly the whole draft, and develop the much-needed logical and scientific cohesion between your individual components for your paper to be considered for publication in RS.
Author Response
We would like to reiterate that we have considered the comments that were most relevant and fitting and have edited and structurally modified the draft if there are major concerns as noted in your message- we would much appreciate specific comments about the same- the general comment about missing a logical structure and natural and scientific cohesion and legitimacy between the individual parts is not very helpful to capture the mind of reviewer or for that matter what is intended. We would like to strongly disagree with the latter part of the comment- as a statement is made without provisions. Note that the Abstract of the paper outlines the structure “The study is divided into three sections (summarised in the below bullet points) ” and the content thereafter follow that sequence. We can take further note of the review comments if they are specific and explicit.
• the application of geospatial systems to assess the water-migration interlinkages- general assessment- water crisis as a direct driver of migration
• outlining a mixed toolset framework that combines RS and Geographic Information System and is applied to three case studies and geospatial outputs analyzed in tandem with climate variables and socio-economic data
• Provides an overview of conflicts and migration aspects by employing the Global Conflict Risk Index (GCRI) projected in a GIS medium to serve as a proxy indicator to show how environmental, political and social dimension act together to steer a conflict- that in turn can lead (acting as an indirect driver) to migration
Reviewer 3 Report
Given the comments I made in the first review of the article that the methodology and analysis proposed in it are more typical of another type of journal, the authors state that the article could be included in a special volume related to the SDG. Revised the calls for special issues of the journal does not appear any with this theme. Likewise, this reviewer does not know what the authors know about this volume related to SDG in Remote Sensing journal. The authors have basically contributed in this second version of the revised article all the editorial comments made by another reviewer and providing the dates of the LandSat scenes used for the multitemporal study of the study areas. In view of the dates provided, it can be concluded that: • The choice of the scenes used is not justified except for their availability. • Their dates are not analyzed in relation to the variation of water reservoirs. • In some cases the distances between dates are small and in others it is large, almost a decade. It is striking that there are no other intermediate scenes. • Trends and seasonality of temperature and rain data obtained from the Climate portal are not minimally analyzed. These facts make me conclude that the analyzes can be sexed, since they do not contemplate the typical cycles that the climate and this type of processes usually have and therefore the conclusions are not completely valid. If the authors seek only to publish the methodology used to integrate data from different sources, including remote sensing scenes for this type of studies in the field of SDGs, with which I agree, no substantial and validated contribution is made. I maintain my opinion expressed in the first review that the article does not fit in this journal.Author Response
I am note sure about the depth of this review comments. It is quite confusing “. Revised the calls for special issues of the journal does not appear any with this theme”, not sure what this means. We have addressed all the thematic comments that was suggested by the reviewer as he/she agrees.
Note that this submission is a unique attempt to apply geospatial approach for migration assessment and focus of the synthesis is on the approach (including methodology) to integrate data from different sources and corelating it with the socioeconomic impacts (reflected in selected SDGs). We note that our submission matched the keywords [SDGs, remote sensing, agriculture. environmental monitoring and land evaluation] outlined for the special volume and the text fits with the content eligibility outlined for the special volume. I would take the focus of the reviewer (https://www.mdpi.com/journal/remotesensing/special_issues/M_SDG_ calls for papers focusing on environmental sustainability , actions of land users such as farmers, urban developments. The environmental changes are detection using global Earth observation systems and remote sensing technologies. Commentary on the challenges of acting upon these SDGs, and facilitation of information flows to make informed choices. The special issue on "Monitoring Sustainable Development Goals" to support the achievement of SDGs and help the involved stakeholders. The submission of research papers includes the exploitation of geospatial data. The contributions that can cover multi-disciplinary aspects of sustainable development, and social resilience to environmental changes, the application and development of geospatial techniques (methods) for the monitoring of land-use/land-changes and the socio-economical impacts of global climate changes. Out submission focused on the relatively under-reflected socio-economical impacts of water and climate crisis- human migration and ticks many boxes outlined as a criterion of submission of the special volume – I leave the rest for the editorial committee to decide the fit with the special issue etc.. To our understanding it does quite clearly and evidently. I am not clear with/on some parts of the review comments to detail the response beyond this point.